# The Effects of Silencing PTX3 on the Proteome of Human Endothelial Cells

**DOI:** 10.3390/ijms232113487

**Published:** 2022-11-03

**Authors:** Cristina Banfi, Maura Brioschi, Lucia M. Vicentini, Maria Grazia Cattaneo

**Affiliations:** 1Unit of Functional Proteomics, Metabolomics and Network Analysis, Centro Cardiologico Monzino, IRCCS, 20138 Milano, Italy; 2Department of Medical Biotechnology and Translational Medicine, Università degli Studi di Milano, 20129 Milano, Italy

**Keywords:** long pentraxin PTX3, inflammation, oxidative stress, endothelial cells, endothelial dysfunction, proteome

## Abstract

The human long pentraxin PTX3 has complex regulatory roles at the crossroad of innate immunity, inflammation, and tissue repair. PTX3 can be produced by various cell types, including vascular endothelial cells (ECs), in response to pro-inflammatory cytokines or bacterial molecules. PTX3 has also been involved in the regulation of cardiovascular biology, even if ambiguous results have been so far provided in both preclinical and clinical research. In this study, we compared the proteomic profiles of human ECs (human umbilical vein ECs, HUVECs), focusing on differentially expressed proteins between the control and PTX3-silenced ECs. We identified 19 proteins that were more abundant in the proteome of control ECs and 23 proteins that were more expressed in PTX3-silenced cells. Among the latter, proteins with multifunctional roles in angiogenesis, oxidative stress, and inflammation were found, and were further validated by assessing their mRNAs with RT-qPCR. Nevertheless, the knock down of PTX3 did not affect in vitro angiogenesis. On the contrary, the lack of the protein induced an increase in pro-inflammatory markers and a shift to the more oxidative profile of PTX3-deficient ECs. Altogether, our results support the idea of a protective function for PTX3 in the control of endothelial homeostasis, and more generally, in cardiovascular biology.

## 1. Introduction

The endothelium, i.e., the continuous cell monolayer that lines blood vessels, plays a critical role in the regulation of vascular tone and in the establishment of an antioxidant, anti-inflammatory, and antithrombotic interface between blood and tissues [1]. A reduced vasodilatory tone, and more broadly, any alteration that affects the homeostatic function of the endothelium, including inflammation and oxidative stress, trigger endothelial dysfunction (ED) [1,2]. More recently, changes in endothelial cell (EC) metabolism and endothelial-to-mesenchymal transition (EndMT) have also been implicated in ED [3,4]. Dysfunction of the vascular endothelium has been related to the initiation and progression of various human disorders, including cardiovascular, metabolic, and emerging infectious diseases [5,6,7].

Inflammation and oxidative stress play an important role in EC activation at the onset of ED [2]. However, increasing evidence suggests that, in addition to other physiologic processes, ECs themselves may act as functional contributors in innate and adaptive immune responses. In this way, ECs behave as a novel type of immune cells that are able to secrete cytokines/chemokines, to recognize pathogen- and damage-associated molecular patterns and immune checkpoints, and to perform phagocytosis [8]. 

In this *scenario*, the role of pentraxins, a superfamily of evolutionarily conserved molecules with multi-functional roles in innate immunity, inflammation, and tissue repair, is highly relevant [9,10]. Prototypes of this family include the short pentraxins C-reactive protein (CRP) and serum amyloid P component (SAP), and the more recently identified long pentraxin PTX3. PTX3 can be produced locally by various cell types, including vascular ECs, in response to pro-inflammatory cytokines or bacterial moieties. PTX3 has also, however, been detected in the secretome of unstimulated ECs [11,12,13]. An important role of PTX3 in the clearance of apoptotic bodies and resolution of inflammation has been proposed in different cell types and tissues, including ECs [13,14,15,16,17,18]. In addition, PTX3 has been involved in the regulation of vascular integrity and cardiovascular biology, although contrasting results have been so far provided either in preclinical or clinical research [19,20,21]. A deeper understanding of the biological functions of PTX3 in ECs, and possibly of the involved molecular mechanisms, may help to shed light on these conflicting outcomes.

For that purpose, in this study, we have compared the proteomic profiles of human ECs isolated from umbilical cords (human umbilical vein ECs, HUVECs), focusing on the differentially expressed proteins between the control and PTX3-silenced ECs. We concentrated our attention on the proteins involved in the regulation of angiogenesis, inflammation, and oxidative stress to evaluate whether the absence of PTX3 was able to modify in vitro angiogenesis and the inflammatory and redox profiles of ECs.

## 2. Results

### 2.1. Proteomic Analysis of PTX3-Silenced ECs

We analyzed the proteomic profile of HUVECs (abbreviated as ECs) collected after 48 h of transfection with 5 nM of the control or PTX3 siRNA. The ability of this protocol to efficiently knock-down PTX3 expression has been shown in our previous paper, where it was also demonstrated that PTX3 silencing did not affect the metabolic, proliferative, or adherent properties of ECs [13]. Proteins expressed in the control and PTX3-silenced ECs were identified and quantified by LC-MS^E^ analysis. We observed a decrease in the expression of 19 proteins (Table 1) and an increase in the levels of 23 proteins (Table 2) in PTX3-silenced ECs in comparison, to control ECs.

Gene ontology (GO) analysis, performed with STRING on the panel of proteins that increased in number in PTX3-silenced ECs, showed a significant enrichment in the biological process categories of GO terms related to mRNA splicing via spliceosomes (*p* = 0.0128) and regulation of angiogenesis (*p* = 0.0254) (Figure 1 and Appendix A). Of note, the enhanced amounts of the angiogenesis-related proteins ephrin type-A receptor 2 (EPHA2) and endoglin (ENG) were validated at the mRNA level, where an increase of 1.7- and 1.9-fold, respectively, was found in PTX3-silenced ECs in comparison to control ECs (Figure 2a,b). 

### 2.2. The Effect of Silencing PTX3 on In Vitro Angiogenesis

The GO analysis reported in Figure 1 showed enrichment in the GO terms involved in the regulation of angiogenesis in PTX3-silenced ECs. In fact, in addition to EPHA2 and ENG, an increase in the integrin α-5 (ITGA5) was observed in the proteome of silenced ECs (Table 2). Hence, we analyzed whether the knock-down of PTX3 impacted the distinctive property of ECs to undergo in vitro angiogenesis. We first measured the expression of the two main proteins involved in the regulation of angiogenesis, i.e., the kinase insert domain receptor (KDR), also known as the vascular endothelial growth factor receptor-2 (VEGFR-2), and the endothelial nitric oxide synthase (eNOS) [22]. The expression of both of the proteins was unchanged in the absence of PTX3, in comparison to control ECs (Figure 3a,b), and accordingly, in vitro sprouting of spheroids from the control or PTX3-silenced ECs did not show any difference in any of the parameters tested (Figure 3c–f). Therefore, the retained ability to undergo in vitro angiogenesis, as well as the described preservation of the metabolic, proliferative and adhesive properties of PTX3-silenced ECs [13], suggest that the modification of angiogenesis-related GO terms following the known-down of PTX3 does not broadly affect the in vitro angiogenic process in cultured ECs.

### 2.3. The Effect of Silencing PTX3 on the Inflammatory Profile of ECs

The key role of PTX3 at the crossroad of immune and inflammatory responses [9,10], as well as the proposed involvement of EPHA2 and ENG in vascular injuries and inflammation [23,24,25,26,27], prompted us to investigate whether the loss of PTX3 might influence the expression of pro-inflammatory genes and cytokines in ECs. We started by measuring the gene expression of the prostaglandin-endoperoxide synthases (PTGS), also known as cyclooxygenases (COX), responsible for the synthesis of prostanoids involved in the inflammatory response [28]. Notably, a significant increase in the expression of the inducible COX-2 enzyme (by 4.5 ± 0.6 fold) was observed in PTX3-silenced ECs in comparison to control ECs (Figure 4a). Regarding variance, the expression of the constitutive COX-1 enzyme was unchanged between the control and silenced ECs. Regarding pro-inflammatory cytokines, we observed in PTX3-silenced ECs (i) a significant increase in the interleukin-1 beta (IL-1β) mRNA (by 3.4 ± 0.7 fold) (Figure 4b) and (ii) a rise in the gene expression of the interleukin-6 (IL-6) (by 1.6 ± 0.3 fold); however, this did not reach statistical significance (Figure 4c). These results support the idea of a suppressive role for PTX3 in the control of endothelial inflammation. Nevertheless, the expression of the monocyte chemoattractant protein-1 (MCP-1), a potent monocyte-attracting chemokine [29], was reduced by about 22% in the absence of PTX3 (Figure 4d).

### 2.4. The Effect of Silencing PTX3 on the Redox Status of ECs

Besides inflammation, another critical player in the onset of ED is oxidative stress, which is due to an imbalance between the production and accumulation of ROS and the ability of cells and tissues to detoxify these products [30]. Since EPHA2 and ENG, the proteins that we found to be increased in PTX3-silenced cells, may promote oxidative stress [24,31], we checked whether the lack of PTX3 expression modified not only the inflammatory profile, but also the redox status of ECs. Oxidative stress was tested in control and PTX3-silenced ECs by (i) the direct detection of ROS, e.g., hydrogen peroxide (H_2_O_2_) (Figure 5a–c); (ii) the measurement of the reduced (GSH)-to-oxidized (GSSG) glutathione ratio, where GSH is one of the most important scavengers of ROS, and its ratio with GSSG represents a dynamic balance between oxidants and antioxidants (Figure 5d); (iii) the expression of antioxidant enzymes, such as the heme-oxygenase-1 (HMOX-1) and superoxide dismutase-2 (SOD-2) enzymes (Figure 5e). The intracellular content of H_2_O_2_ was approximately double in PTX3-silenced ECs in comparison to control ECs (53.100 ± 5.444 vs. 27.270 ± 3.103 RLU/µg protein, respectively) (Figure 5a). Likewise, the extracellular levels of H_2_O_2_ measured over time constantly increased (Figure 5b) and the slope value of the interpolated curve was significantly higher (0.022 ± 0.02 vs. 0.014 ± 0.01 slope/min) (Figure 5c) in PTX3-silenced ECs, suggesting a lower scavenging capacity of ECs devoid of PTX3. The presence of an unbalanced redox condition in PTX3-silenced ECs is confirmed by (i) the decrease in the GSH/GSSG ratio (by about 26%) (Figure 5d) and (ii) the enhanced mRNA expression of both HMOX-1 and SOD-2 (by 1.63 ± 0.3 and 1.60 ± 0.1, respectively) (Figure 5e). Collectively, all these results suggest that the expression of PTX3 is required to maintain redox homeostasis in ECs.

## 3. Discussion

The long pentraxin PTX3 is a crucial component of humoral innate immunity produced at inflammatory sites in response to infection or tissue injury [9,10]. However, PTX3 also holds ambiguous properties regarding the regulation of cardiovascular biology due to its ability to modulate angiogenesis, vascular remodeling, and inflammation [20,21,32]. In this study, we compared the proteome of the control and PTX3-silenced human ECs, which showed some differences in the proteins with multifunctional roles, such as angiogenesis, inflammation, and oxidative stress. Nevertheless, the knock-down of PTX3 did not affect sprouting in the 3D assay of in vitro angiogenesis, suggesting that the dysregulation of the identified proteins may be counterbalanced by other factors. On the contrary, the lack of PTX3 induced an increase in inflammatory markers and the shift of PTX3-deficient ECs to a more oxidative profile. Taken together, these results support the idea that PTX3 may possibly carry out a protective function in the endothelium, and more generally, in cardiovascular homeostasis. 

As mentioned above, PTX3 is a component of the humoral arm of innate immunity, and its expression in human tissues is usually associated with inflammation [9,10,32]. However, both pro-inflammatory and inflammation-limiting properties have been reported in preclinical models, unveiling a possible dual role in physio-pathological conditions. Anti-inflammatory cytokines and atheroprotective signals, i.e., interleukin-10 (IL-10) and high-density lipoproteins (HDL), are able to induce PTX3 expression [33,34]. In addition, the deletion of PTX3 increases vascular inflammation and macrophage accumulation at site of atherosclerotic plaque in apolipoprotein E–knockout mice [34,35]. The PTX3-mediated inhibition of the pro-thrombotic effects of fibrinogen and collagen l is responsible for the protective role described in an experimental model of arterial thrombosis [36]. In addition, a cardio-protective effect has also been reported for PTX3 in murine models of acute myocardial infarction, and increased plasma levels and cardiac expressions of IL-6 and generation of ROS have been shown in the ischemic myocardium of PTX3-knock out animals [37,38]. Remarkably, the involvement of PTX3 in the clearance of apoptotic bodies via the execution of efferocytotic processes may also be involved in its anti-inflammatory properties [13]. In fact, phagocytosis of dying cells prevents the release of inflammatory factors, the establishment of inflammation, and the development of chronic inflammatory disorders, such as atherosclerosis [39,40]. Regarding variance, overexpression of PTX3 inhibits nitric oxide (NO) production in either mice or human ECs [41], and the strong expression of PTX3 has been reported in human advanced atherosclerotic lesions [42]. Besides preclinical results, clinical data have so far provided contrasting results on the role of PTX3 in the regulation of vascular integrity and cardiovascular biology [20,21]. PTX3 plasma levels have been found to be significantly elevated in several clinical settings, including acute myocardial infarction, heart failure and cardiac arrest (where they have been related to the extent of tissue damage and risk of mortality), small vessel vasculitis and rheumatoid arthritis, chronic kidney disease and preeclampsia [21,43]. However, PTX3 has mainly been investigated as a potential biomarker, and we have limited knowledge on its functional involvement in these pathologies. To date, it is not possible to differentiate between whether circulating PTX3 actively sustains the inflammatory process or merely acts as a viewer or exerts a protective physiological response that is correlated with disease extent and severity. 

Our data, which show the development of a pro-inflammatory and more oxidative profile in PTX3-deficient ECs, suggest the requirement of PTX3 for the control of endothelial inflammation and oxidative stress, thus favoring its role as a vascular protective factor. Further studies are needed to reveal the downstream mechanism(*s*) responsible for these effects. In this paper, we proposed two proteins as possible mediators of the observed phenotype, i.e., EPHA2 and ENG, both endowed with pro-inflammatory and oxidant properties [23,24,25,26,27]. Overall, minor changes in the number of differentially expressed proteins (about 20 proteins, corresponding to ~2% of all proteins) have been observed the between control and PTX3-silenced ECs. In addition, the extent of variations in protein levels was rather small (on average, 1.3-fold, ranging from 1.2 to 1.7). However, even small differences in sets of protein can result in functional changes, especially in unstimulated native cells not exposed to treatments or stressors. In addition, post-translational modifications (PTMs) can modulate protein activities and the presence of isoforms derived from alternative splicing can also contribute to the observed biological effects. Therefore, it must be noted that unknown modified proteins or isoforms could play a part in defining the inflammatory and redox profiles shown by ECs in the absence of PTX3, even though the bottom-up LC-MS^E^ approach used in this study hampers the detection of PTMs and isoforms [29]. As a result, the enhanced expression of proteins/enzymes involved in the regulation of the splicing process in PTX3-silenced ECs is highly relevant. Interestingly, most of the genes involved in the triggering of inflammation and oxidative stress in PTX3-deficient ECs are regulated at the transcriptional level, and disease-specific gene expression profiles have been associated with atherosclerosis. More recently, single-cell sequencing revealed cell-specific atherosclerotic transcriptomes [44], and several studies highlighted the critical role of epigenetic mechanisms in ED [45]. Therefore, it will be critical to characterize the transcriptomic and epigenomic profiles of PTX3-deficient ECs to further study the pathways responsible for the shift toward a dysfunctional endothelium.

Irrespective of the underlying mechanisms, the lack of PTX3 induced an increase in the expression of the archetypal pro-inflammatory cytokine IL-1β in ECs [46,47]. IL-1β can induce its own gene expression, providing a positive feedback amplification loop for enhancing its levels at the site of inflammation. In addition, IL-1β boosts the production of IL-6 that, in turn, drives the expression of atherothrombosis mediators [48]. Notably, IL-1β also stimulates the synthesis of pro-inflammatory prostaglandins through the production of the inducible COX-2 enzyme [28]. Accordingly, the increase in IL-1β is accompanied by a rise in both IL-6 and COX-2 expression in PTX3-silenced ECs. Of note, some inflammatory mediators, including IL-1β, can convert ECs into mesenchymal-like cells through the EndMT process. Consequently, ECs lose their typical cobblestone morphology, as well as their expression of endothelial markers and properties. In this way, EndMT contributes to the pathogenesis of different disorders, such as ED and atherosclerosis, fibrotic diseases, and cancer [4,49]. Therefore, it will be noteworthy to evaluate either the morphology or the expression of EC markers, cell–cell adhesion molecules and mesenchymal factors in PTX3-silenced ECs to reveal the potential activation of the EndMT process. Significantly, the down-regulation of PTX3 expression has recently been demonstrated in patients with idiopathic pulmonary fibrosis [18]. 

In addition to inflammation, the loss of PTX3 was associated with the accumulation of ROS and reduced scavenging activity, which is consistent with the decrease in the GSH/GSSG ratio, as GSH is one of the most important intracellular scavengers. These results are also in accordance with the reduced expression in PTX3-silenced ECs of the endogenous antioxidant enzyme glutathione S-transferase P (GSTP1) [50]. Furthermore, the increased expression of the antioxidant enzymes HMOX-1 and SOD-2 [51,52] pointed to the detoxification of harmful metabolites as an adaptive response mechanism in PTX3-silenced ECs. In addition to the impairment of NO-dependent vasodilation and the development of chronic low-grade inflammation, the induction of oxidative stress is also a hallmark of ED [2,30]. Thus, the concurrent alterations in the inflammatory and redox profiles induced by the absence of PTX3 further support the hypothesis of a protective role for this protein in ECs, and more generally, in vascular homeostasis.

The main limitation of our study is the use of ECs from a unique source, the human umbilical vein. Indeed, organ specificity and the function of the endothelium, and subsequently of ECs, have been demonstrated [53]. From this perspective, HUVECs derive from a rather unique type of tissue with exclusive properties (it is, among others, the first interface between the fetus and mother). Despite this unicity, HUVECs still represent the main model for studying in vitro properties of the endothelium and ECs [54]. HUVECs undergo in vitro angiogenesis, produce NO, respond to shear stress, hyperglycemia, and inflammatory stimuli, thus recapitulating human disease pathophysiology and supporting their use as a valuable model for basic research on the endothelium. Transcriptomic analysis has recently shown that the expression of more than 250 genes in male and female HUVECs is concordantly maintained in adult human aortic ECs [55], further strengthening the translational significance of using HUVECs in basic research. Nevertheless, future studies with ECs derived from different organs and/or vascular beds would be very helpful to confirm the general role of PTX3 in the regulation of inflammatory and oxidative properties of the endothelium. Among others, it should be of great interest to study brain microvascular EC (BMVEC)-derived PTX3 and its role in the protective effect observed in neurons after seizures or strokes [56]. In addition, the upregulation of PTX3 has been shown in the brain of mice subjected to a traumatic brain injury (TBI) [57]. The involvement of BMVEC-derived PTX3 in the onset and/or in the resolution of the inflammatory process following TBI will also help us to expand our knowledge of the detrimental or beneficial effects of PTX3 in different models of central or peripheral tissue injuries and inflammation.

In conclusion, our findings show that the lack of PTX3 triggers inflammation and oxidative stress in human vascular ECs, namely HUVECs, thus supporting its potential regulatory role as an anti-inflammatory and anti-atherogenic molecule. Although still debated, pleiotropic functions of PTX3 in vascular biology are of growing interest. Substantial evidence suggests the dual role of PTX3 as a modulator or amplifier of the innate immune response, and the biological consequences of its production and activity may depend on a fine balance between the cell/tissue of origin and environmental signals. Knowledge of how PTX3 acts at the crossroad between pro- and anti-inflammatory signals at the vascular bed may represent a major challenge in the identification of novel pathogenetic mechanisms and pharmacological targets for the prevention and treatment of ED. Likewise, awareness of the capacity of ECs to adjust themselves to inflammatory/immune responses, and the capacity of the involved molecular pathways, may further reveal new preventive and therapeutic approaches to reduce inflammation at the onset of atherosclerosis and cardiovascular disease.

## 4. Materials and Methods

### 4.1. Cell Cultures

Human umbilical vein endothelial cells (HUVECs, abbreviated as ECs) were either isolated from umbilical cords by digestion with collagenase [58] or purchased from Lonza (Bend, OR, USA), and cultured on 0.1% gelatin-coated surfaces. These former cells were routinely grown in 199 media (M199), supplemented with 20% fetal bovine serum (FBS), 25 µg/mL endothelial cell growth factor (ECGF) and 50 µg/mL heparin. The latter were grown in Endothelial Growth Medium (EGM™-2), as indicated by the provider (Lonza, Bend, OR, USA). Both the ECs were derived from pooled donors to minimize variability and used in steps 1–5.

### 4.2. Label-Free Mass Spectrometry (LC-MS^E^) Analysis

Cell pellets were dissolved in 25 mmol/L NH_4_HCO_3_ containing 0.1% RapiGest (Waters Corporation, Milford, MA, USA), sonicated, and centrifuged at 13,000× *g* for 10 min. After 15 min of incubation at 80 °C, proteins were reduced with 5 mmol/L DTT at 60 °C for 15 min, and carbamidomethylated with 10 mmol/L iodoacetamide for 30 min at room temperature in darkness. Digestion was performed with sequencing grade trypsin (Promega, Milan, Italy) (1 µg every 50 µg of proteins) overnight at 37 °C. After digestion, 2% TFA was added to hydrolyze RapiGest and inactivate trypsin. Tryptic peptides were used for label-free mass spectrometry analysis, LC-MS^E^, which was performed on a hybrid quadrupole time-of-flight mass spectrometer (SYNAPT-XS, Waters Corporation, Milford, MA, USA), coupled with an UPLC Mclass system and equipped with a nanosource (Waters Corporation). Samples were injected into a Symmetry C18 nanoACQUITY trap column, 100 Å, 5 μm, 180 μm × 2 cm (Waters Corporation, Milford, MA, USA), and subsequently directed to the analytical column HSS T3 C18, 100 Å, 1.7 μm, 75 μm × 150 mm (Waters Corporation, Milford, MA, USA), for elution at a flow rate of 300 nL/min by increasing the organic solvent B concentration from 3 to 40% over 90 min, using 0.1% v/v formic acid in water as reversed phase solvent A, and 0.1% v/v formic acid in acetonitrile as reversed phase solvent B. All of the analyses were performed in triplicate and analyzed by LC-MS^E^ as previously detailed [59], with some modifications in ion mobility-enhanced data-independent acquisition (IMS-DIA). In particular, in the low-energy MS mode, the data were collected at a constant collision energy of 6 eV, while in the high-energy mode, fragmentation was achieved by applying drift time-specific collision energies [60]. Statistical analysis was performed by means of Progenesis QIP v 4.1 (Nonlinear Dynamics) using a Uniprot human protein sequence database (v2020-7) and data were deposited in the ProteomeXchange Consortium via the PRIDE [61] partner repository. 

### 4.3. Gene Ontology Analysis

The Search Tool for the Retrieval of Interacting Genes/Proteins (STRING 11.5) database [62] was used to analyze the network of modulated proteins, as previously described [63]. Briefly, we employed the enrichment widget of STRING to identify enriched gene ontology (GO) terms in the biological process, molecular function, or cellular component categories (*p* value cut-off of <0.05).

### 4.4. Small Interfering RNA (siRNA) Transfection

To silence PTX3 expression, ECs were transfected with the Hs_PTX3_1 FlexiTube siRNA duplexes against human PTX3 (SI00695947, Qiagen, Milano, Italy). The AllStars Negative Control FlexiTube siRNA (Qiagen, Milano, Italy) was used as the control. Both siRNAs were individually transfected at a 5 nM concentration using the PepMute transfection reagent according to the manufacturer’ instructions (Signa Gen Laboratories, Frederick, MD, USA). All the experiments were performed after 48 h of transfection. The ability of siRNA to knockdown PTX3 expression was measured by RT-qPCR. The percentage of target mRNA reduction was ≥ 85% in all the experiments [13]. For sprouting assays, ECs were transfected 24 h before plating to form spheroids [64,65]. 

### 4.5. Total RNA Extraction and Reverse Transcription and Quantitative Real-Time PCR (RT-qPCR)

Total RNA was extracted using the *Quick*-RNA Miniprep Kit and accompanying Spin-Away™ Filter (Zymo Research, Irvine, CA, USA). To avoid DNA contamination of samples, 15 min of column incubation with DNase I was carried out (Zymo Research, Irvine, CA, USA). For the quantitative analysis of PTGS-2/COX-2, HMOX-1, and SOD-2 gene expression, reverse transcription was performed with the All-In-One 5X RT MasterMix (Applied Biological Materials), and the QuantStudio™ 5 Real-Time PCR System (Thermo Fisher Scientific, Rodano, Italy) was used for qPCR. Target sequences were amplified from 50 ng of cDNA using the validated TaqMan™ Primer and Probe assays (Thermo Fisher Scientific, Rodano, Italy) for the human PTGS-2/COX-2 (Hs00153133_m1), HMOX-1 (Hs01110250_m1), SOD-2 (Hs00167309_m1), and for the endogenous control 18 S (Hs99999901_s1). 

For the quantitative analyses of PTX3, EPHA2, ENG, PTGS-1/COX-1, IL-1β, IL-6, and MCP-1, total RNA extracted, as defined above, was reverse transcribed as previously described [66]. RT-qPCR was performed in triplicate with 2.5 μL of cDNA incubated in 22.5 μL of IQ Supermix that contained primers and SYBRGreen fluorescence dye (Bio-Rad Laboratories, Milan, Italy), using the iCycler Optical System (Bio-Rad Laboratories, Milan, Italy). The sequences of the primers are reported in Appendix A. GAPDH was used as the endogenous control. All the experiments were performed in triplicate and for the calculation of results, the 2^−ΔΔCt^ method was used.

### 4.6. Immunoblotting

ECs were lysed in Laemli’s sample buffer, and proteins were quantified by the BCA assay (Thermo Fisher Scientific, Rodano, Italy). Next, 20–30 µg of the total proteins were loaded on 4–20% polyacrylamide precast gels (Criterion TGX Stain-Free precast gels; Bio-Rad Laboratories, Segrate, Italy). Before transfer, short photoactivation with UV light made the proteins fluorescent and allowed their visualization, permitting us to obtain quantitative immunoblots by normalizing the bands of interest to the total proteins loaded in each lane. The gels were then transferred onto a nitrocellulose membrane using a Trans-Blot Turbo System™ and Transfer pack™ (Bio-Rad Laboratories, Segrate, Italy). Membranes were probed with the primary and secondary antibodies listed in the “Reagent and Antibodies” section, according to the manufacturer’ instructions. Results were acquired with the Bio-Rad ChemiDoc XRS Imaging system and analyzed with the Image Lab software (Bio-Rad Laboratories, Segrate, Italy). The Stain-Free total protein measurement was used as a loading control, as it is more reliable than housekeeping proteins. Full-length unedited blots are shown in Appendix A.

### 4.7. In Vitro Angiogenesis

To measure the ability of ECs to undergo in vitro angiogenesis, the three-dimensional (3D) spheroid sprouting assay was used. This assay is a well-established and robust method that allowed us to study the influence of genetic alterations or pharmacological compounds on capillary-like tube formations in primary cultured ECs [65,67,68]. Briefly, ECs (1.000 cells/well in low-attachment 96-well round bottom plates) were incubated overnight in the standard growth medium containing 20% carboxymethylcellulose to form spheroids. Spheroids were then collected and embedded into collagen gels, as previously described [69]. Images were acquired 24 h later at 10× magnification with an Olympus U-CMAD3 microscope equipped with an Olympus digital camera. In-gel angiogenesis was quantified by measuring the cumulative length, the average length, and the number of all the capillary sprouts that originated from individual spheroids, using the National Institute of Health (NIH) Image J software package. A minimum of 20 randomly selected spheroids *per* experimental group were measured in each experiment. 

### 4.8. Measurement of Hydrogen Peroxide and GSH/GSSG Ratio

ECs were plated at a density of 2.0 × 10^4^ cells/well in 0.1% gelatin-coated 96-well microplates. After overnight incubation, media were changed to HBSS buffer (Hepes 25 mM pH 7.4, NaCl 120 mM, KCl 5.4 mM, CaCl_2_ 1.8 mM, NaHCO_3_ 25 mM; glucose 15 mM), and intra- and extra-cellular H_2_O_2_ levels were measured with a commercially available kit (ROS-Glo H_2_O_2_ Assay, Promega, Milano, Italy), following the manufacturer’s instructions. Similarly, the GSH/GSSG ratio was calculated by the total glutathione and GSSG levels detected with the GSH/GSSG-Glo™ Assay (Promega, Milano, Italy). Luminescence was measured with the GloMax™ Discover Microplate Reader (Promega, Milano, Italy) and normalized to the protein content of each well quantified with the BCA assay (Thermo Fisher Scientific, Rodano, Italy). 

### 4.9. Statistical Procedures

Unless otherwise indicated, data are expressed as mean ± s.e.m. of at least 3 independent experiments performed on different cell preparations. Statistical analysis was carried out by unpaired Student’s *t*-test. *p*-values of <0.05 were considered significant. All the analyses were performed using the GraphPad Prism software (version 9.4.1). 

### 4.10. Reagents and Antibodies

All tissue culture reagents were obtained from Euroclone, except ECGS and heparin (Sigma Aldrich) and EGM-2 (Lonza, Bend, OR, USA). Carboxymethylcellulose (M0512) was obtained from Merck KGaA, Darmstadt, Germany and type 1 rat tail collagen from Serva Electrophoresis GmbH, Heidelberg, Germany. The primary antibodies used were rabbit polyclonal anti-KDR (Santa Cruz Biotechnology, Heidelberg, Germany, sc-504), and mouse monoclonal anti-eNOS (BD Transduction Laboratories, La Jolla, CA, USA, 610296). Horseradish peroxidase (HRP)-conjugated secondary antibodies were obtained from Dako (Agilent, Santa Clara, CA, USA, #P0399 and #P0260 for swine anti-rabbit and rabbit anti-mouse antibodies, respectively). Proteins in Western blot were detected by the LiteAblot Turbo Extra-Sensitive Chemiluminescent Substrate (Euroclone, Pero, Italy). 

## Figures and Tables

**Figure 1 ijms-23-13487-f001:**
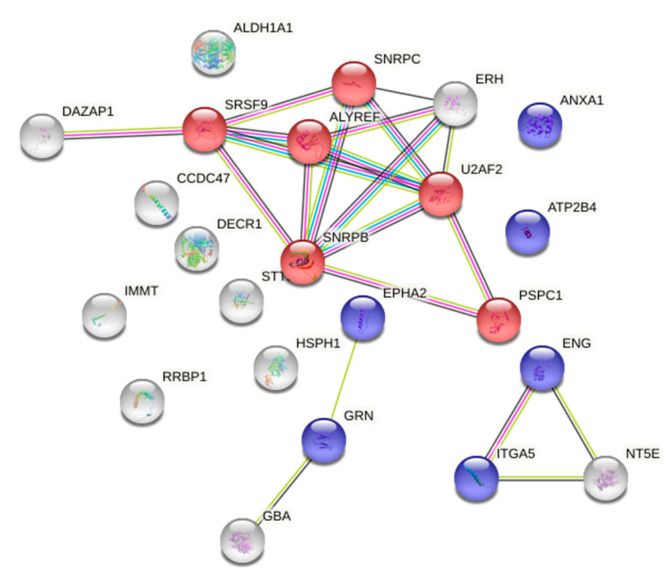
Gene ontology (GO) analysis of the more expressed proteins in PTX3-silenced ECs (as shown in Table 2). String network generated with these proteins that highlights the following enriched biological processes: red, mRNA splicing; blue, regulation of angiogenesis.

**Figure 2 ijms-23-13487-f002:**
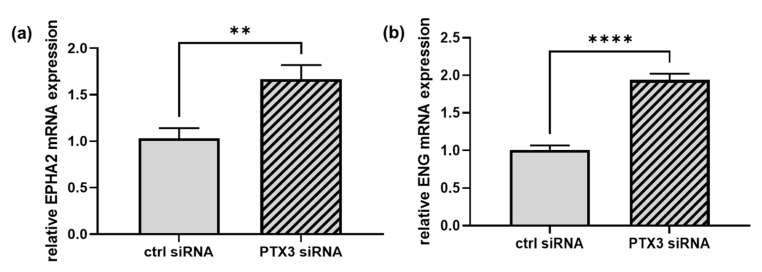
RT-qPCRs were performed on mRNAs prepared from ECs transfected for 48 h with control siRNA (CTRL siRNA, solid bars) or PTX3 siRNA (diagonal bars). EPHA2 (**a**) and ENG (**b**) mRNAs were normalized to the housekeeping gene GAPDH. (**a**), ** *p* < 0.001, *n* = 6. (**b**) **** *p* < 0.0001, *n* = 6.

**Figure 3 ijms-23-13487-f003:**
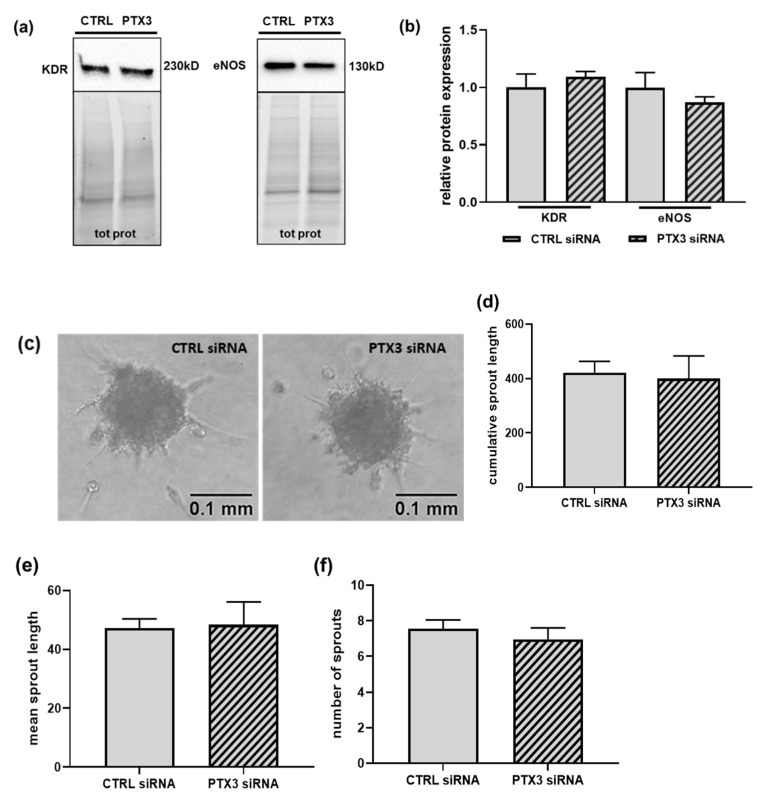
The effect of silencing PTX3 on in vitro angiogenesis. ECs were transfected for 48 h with control siRNA (CTRL siRNA, solid bars) or PTX3 siRNA (diagonal bars). (**a**) Representative immunoblots of KDR (left panel) and eNOS (right panel) in lysates from CTRL or PTX3-silenced ECs normalized to the total protein content. Densitometric quantification is provided in (**b**). KDR, *p* = 0.498, *n* = 4; eNOS, *p* = 0.405, *n* = 3. (**c**) Representative images of spheroids from control ECs (CTRL siRNA) (left panel) or PTX3-silenced ECs (PTX3 siRNA) (right panel) embedded in collagen gels, as described in the “Materials and Methods” section. Photographs were taken 24 h later. Scale bar, 0.1 mm. Quantification of in vitro angiogenesis by measurement of the cumulative length, mean length, and number of all the processes in individual spheroids formed by ECs transfected with CTRL siRNA (solid bars) or PTX3 siRNA (diagonal bars) are shown in (**d**–**f**), respectively. Data are the mean ± SEM of 20 randomly selected spheroids for experimental group.

**Figure 4 ijms-23-13487-f004:**
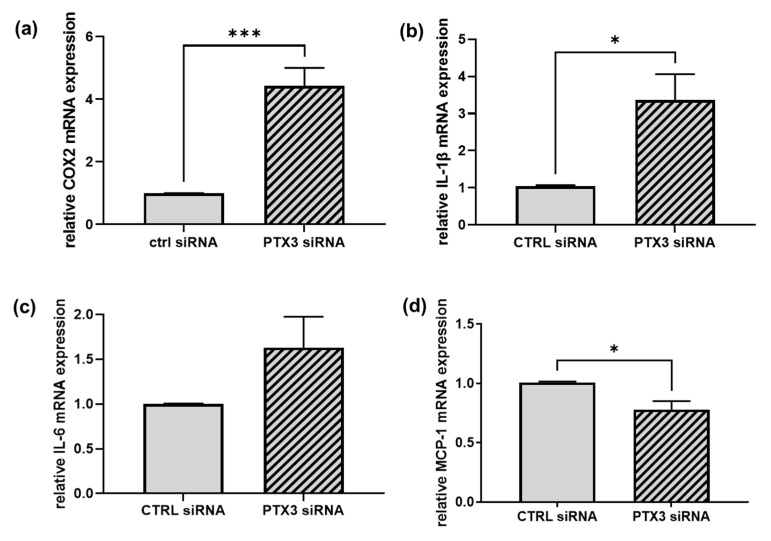
The effect of silencing PTX3 on the inflammatory profile of ECs. RT-qPCRs were performed on mRNAs prepared from ECs transfected for 48 h with control siRNA (CTRL siRNA, solid bars) or PTX3 siRNA (diagonal bars). In (**a**), COX-2 mRNA was normalized to the housekeeping gene 18S; in (**b**–**d**), IL-1β, IL-6 and MCP-1 mRNAs were normalized to GAPDH. (**a**), *** *p* < 0.001, *n* = 4. (**b**) * *p* < 0.05, *n* = 3. (**c**) *p* = 0.143, *n* = 3. (**d**) * *p* < 0.05, *n* = 4.

**Figure 5 ijms-23-13487-f005:**
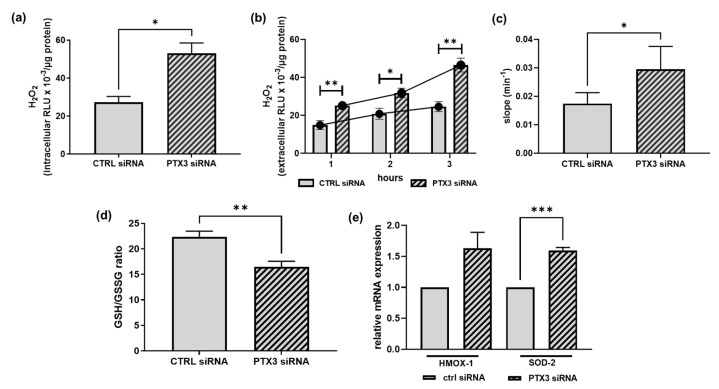
The effect of silencing PTX3 on the redox state of ECs. H_2_O_2_ was measured by the ROS-Glo™ H_2_O_2_ assay in the control (CTRL siRNA, solid bars) or PTX3-silenced ECs (PTX3 siRNA, diagonal bars). Results are expressed as relative luminescence units (RLU) normalized to the protein content of each well. In (**a**), intracellular H_2_O_2_ was measured in cell lysates after 3 h in HBSS. * *p* < 0.05, *n* = 4. In (**b**), extracellular H_2_O_2_ was measured at the indicated times following the manufacturer’s instructions. * *p* < 0.05, ** *p* < 0.01, *n* = 4. Curves interpolated between the mean values of each bar are shown. In (**c**), slope values of the curves shown in (**b**) were compared by linear regression analysis. * *p* < 0.05, *n* = 4. In (**d**), the GSH/GSSG ratio was measured by the GSH/GSSG-Glo™ Assay in the control (CTRL siRNA, solid bars) or PTX3-silenced ECs (PTX3 siRNA, diagonal bars). ** *p* < 0.01, *n* = 4. In (**e**), HMOX-1 and SOD-2 mRNA expression were measured by RT-qPCR in the control (CTRL siRNA, solid bars) or PTX3-silenced ECs (PTX3 siRNA, diagonal bars) and normalized to the housekeeping gene 18S. *** *p* < 0.0001, *n* = 4.

**Table 1 ijms-23-13487-t001:** List of the proteins identified by LC-MS^E^ analysis and reduced by PTX3 silencing in the EC proteome.

Accession	Description	Peptide Count/Unique Peptides	Score	*p* Value	Fold Change
P09211	Glutathione S-transferase P, GSTP1	10/9	106	0.0022	1.21
P60660	Myosin light polypeptide 6, MYL6	8/8	77	0.0021	1.22
P08708	40S ribosomal protein S17, RPS17	2/2	22	0.0023	1.23
Q86VZ2	WD repeat-containing protein, WDR5B	2/2	13	0.0003	1.25
P12821	Angiotensin-converting enzyme, ACE	2/2	11	0.0063	1.27
Q9BUJ2	Heterogeneous nuclear ribonucleoprotein U-like protein 1, HNRNPUL1	6/5	39	0.0002	1.28
Q7Z406	Myosin-14, MYH14	16/4	134	0.0001	1.30
P30626	Sorcin, SRI	4/4	33	2.49 × 10^−5^	1.31
P25787	Proteasome subunit alpha type 2, PSMA2	3/3	20	0.0002	1.31
Q9NUV9	GTPase IMAP family member 4, GIMAP4	3/3	18	0.0005	1.33
Q96AY3	Peptidyl-prolyl cis-trans isomerase, FKBP10	3/3	17	0.0006	1.34
P49748	Very long-chain specific acyl-CoA dehydrogenase_ mitochondrial, ACADVL	2/2	10	0.0036	1.34
P67936	Tropomyosin alpha-4 chain, TPM4	11/3	107	2.10 × 10^−5^	1.35
P60903	Protein S100-A10, S100A10	2/2	21	1.24 × 10^−6^	1.41
P20290	Transcription factor BTF3, BTF3	2/2	11	0.0005	1.48
Q99439	Calponin-2, CNN2	2/2	13	5.03 × 10^−5^	1.51
P52943	Cysteine-rich protein 2, CRIP2	2/2	14	0.0001	1.52
Q6DD88	Atlastin-3, ATL3	5/5	27	7.54 × 10^−5^	1.53
P62841	40S ribosomal protein S15, RPS15	2/2	19	0.0002	1.73

**Table 2 ijms-23-13487-t002:** List of the proteins identified by LC-MS^E^ analysis and increased by PTX3 silencing in the EC proteome.

Accession	Description	Peptide Count/Unique Peptides	Score	*p* Value	Fold Change
P04062	Lysosomal acid glucosylceramidase, GBA	2/2	11	0.0083	1.21
Q96A33	Coiled-coil domain-containing protein 47, CCDC47	2/2	10	0.0024	1.22
P08648	Integrin alpha 5, ITGA5	12/11	78	0.0001	1.22
P00352	Retinal dehydrogenase 1, ALDH1A1	18/16	154	2.25 × 10^−5^	1.22
P09234	U1 small nuclear ribonucleoprotein C, SNRPC	2/2	13	0.0049	1.23
Q8TCJ2	Dolichyl-diphosphooligosaccharide-protein glycosyltransferase subunit, STT3B	2/2	10	0.0007	1.24
P04083	Annexin A1, ANXA1	16/16	178	1.63 × 10^−6^	1.25
P26368	Splicing factor U2AF 65 kDa subunit, U2AF2	4/4	26	1.48 × 10^−6^	1.25
P14678	Small nuclear ribonucleoprotein-associated proteins B and B’, SNRPB	4/4	26	2.36 × 10^−5^	1.26
Q96EP5	DAZ-associated protein 1, DAZAP1	2/2	13	0.0084	1.26
Q86V81	THO complex subunit 4, ALYREF	4/4	44	0.0001	1.26
Q13242	Serine/arginine-rich splicing factor 9, SRSF9	2/2	13	0.0012	1.27
P29317	Ephrin type-A receptor 2, EPHA2	2/2	12	0.0041	1.28
P21589	5′-nucleotidase, NT5E	4/3	27	2.22 × 10^−5^	1.30
P84090	Enhancer of rudimentary homolog, ERH	3/3	31	0.0016	1.31
Q8WXF1	Paraspeckle component 1, PSPC1	2/2	11	0.0107	1.32
P17813	Endoglin, ENG	7/5	57	3.53 × 10^−5^	1.33
P28799	Progranulin, GRN	2/2	14	0.0053	1.33
Q16891	MICOS complex subunit MIC60, IMMT	7/7	48	0.0001	1.34
Q9P2E9	Ribosome-binding protein 1, RRBP1	26/26	199	8.82 × 10^−5^	1.37
Q16698	2_4-dienoyl-CoA reductase_ mitochondrial, DECR1	4/4	28	6.23 × 10^−6^	1.38
Q92598	Heat shock protein 105 kDa, HSPH1	4/3	21	7.10 × 10^−5^	1.51
P23634	Plasma membrane calcium-transporting ATPase 4, ATP2B4	2/2	10	0.0001	1.68

## Data Availability

The mass spectrometry proteomics data have been deposited in the ProteomeXchange Consortium via the PRIDE [61] partner repository with the dataset identifier PXD036691 and 10.6019/PXD036691. The other data presented in this study are available within the article and in the attached Appendix A.

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
