# Peer review of "The Effects of Silencing PTX3 on the Proteome of Human Endothelial Cells"

_ijms, 2022, doi:10.3390/ijms232113487_

Round 1

Reviewer 1 Report

This paper entitled “The effect of silencing PTX3 on the proteome of human endo- 2

thelial cells” by Banfi et al. investigated effect of silencing PTX3 on HUVEC.

PTX is an interesting gene. Authors tested the PTX3 effected on HUVEC. Similar relationships between PTX3 and inflammatory makers are reported. The method being used were established. The research is meaningful, but overall novelty is limited.

The overall quality of this work is good. The experiment design is good, and proper reference is conduct. The manuscript is well-prepared. Data is convincing and well discussed. Although the experiment design good and data is convincing, there are some of the concerns will be addressed as follows:

1 Does authors considered using 2 endothelial cell lines to proof these conclusions are general in endothelial cells?

2 Limitation of this study should be addressed.

3 Scale bar is needed in Figure 3C

4 Emails is not necessary expect correspondence.

Author Response

 1 Does authors considered using 2 endothelial cell lines to proof these conclusions are general in endothelial cells?

Response: Thank you to the reviewer for her/his comment. We are aware of the intrinsic limitations due to the use of a single experimental model. A brief subsection on the significance of HUVECs as endothelial paradigm has now been added to the revised version of the manuscript (page 9, lines 295-315). As discussed in that paragraph, organ-specificity and function of endothelium, and subsequently of ECs, have clearly been demonstrated. Nevertheless, despite their unicity, HUVECs still represent the most widely used cellular model for studying in vitro properties of human endothelium and ECs. A PubMed search for “HUVEC/HUVECs” performed in October 2022 retries more than 12.350 published papers in the last 10 years, and umbilical cords and HUVECs are the most common tissue and EC subtype represented in the EndoDB database (Khan et al. Nucleic Acids Research 2019 47, D736 doi: 10.1093/nar/gky997). In fact, HUVEC properties (summarized in Medina‐Leyte et al. Appl. Sci. 2020, 10, 938 doi:10.3390/app10030938) recap the most important features of an ideal in vitro model system. They are inexpensive, easily manipulated, quite reproducible, ethically sound, and recapitulate human disease pathophysiology. Of course, HUVECs suffer of limitations – as well as every other experimental model - but unquestionably offer advantages and benefits very helpful in basic research on endothelium. Notably, we obtained overlapping results by silencing PTX3 either in commercial HUVECs purchased by Lonza or in HUVECs prepared from fresh cords in the labs. However, we are currently set up a different EC model represented by the mouse brain microvascular ECs (BMVECs), and it will be interesting to test in the future whether PTX3 maintains a regulatory role on inflammation and oxidative stress also in this model of microvascular origin.

We have also specified “HUVECs” in the conclusion’s subsection (page 9, line 317).

2 Limitation of this study should be addressed.

Response: Following the suggestion of the reviewer, we have now added a paragraph at the end of the Discussion section (page 9, lines 295-315). In this paragraph, we have mainly discussed the meaning of HUVECs as endothelial model. Other limitations of the study, such as for example the bottom-up approach of the proteomic analysis, have already been included in the Discussion (page 8, lines 247-264) and not covered in this new paragraph.

3 Scale bar is needed in Figure 3C.

Response: Scale bar has been added to Fig. 3C in the revised version of the manuscript.

4 Emails is not necessary expect correspondence.

Response: E-mail addresses have been removed in the revised version of the manuscript.

Reviewer 2 Report

In general,

The manuscript entitled "The effect of silencing PTX3 on the proteome of human endothelial cells" is an interesting comparison the proteomic profiles of human ECs isolated from umbilical cords (Human Umbilical Vein ECs, HUVECs), focusing on the proteins differentially expressed between control and PTX3 silenced ECs.

The Instrument is well described, the Materials and Methods are clear and the Conclusions are valuable and useful for many in the field interested in the analysis.

I remain at your service, greetings.

Author Response

In general, the manuscript entitled "The effect of silencing PTX3 on the proteome of human endothelial cells" is an interesting comparison the proteomic profiles of human ECs isolated from umbilical cords (Human Umbilical Vein ECs, HUVECs), focusing on the proteins differentially expressed between control and PTX3 silenced ECs.

The Instrument is well described, the Materials and Methods are clear, and the Conclusions are valuable and useful for many in the field interested in the analysis.

I remain at your service, greetings.

Thank very much to the reviewer for her/his appreciation.

Round 2

Reviewer 1 Report

As authors have successfully answer all these questions, I suggest to publish in present form.